# Expression Analysis and the Roles of the *Sec1* Gene in Regulating the Composition of Mouse Gut Microbiota

**DOI:** 10.3390/genes13101858

**Published:** 2022-10-14

**Authors:** Zhanshi Ren, Hairui Fan, Shanshen Gu, Haoyu Liu, Zhengchang Wu, Haifei Wang, Wenbin Bao, Shenglong Wu

**Affiliations:** 1Key Laboratory for Animal Genetics, Breeding, Reproduction and Molecular Design of Jiangsu Province, College of Animal Science and Technology, Yangzhou University, Yangzhou 225009, China; 2Joint International Research Laboratory of Agriculture & Agri-Product Safety, Yangzhou University, Yangzhou 225009, China

**Keywords:** *Sec1*, knockout mouse, feces, bacterial flora, 16SrDNA sequencing

## Abstract

The *Sec1* gene encodes galactose 2-L-fucosyltransferase, whereas expression during development of the *Sec1* gene mouse and its effect on the composition of the gut microbiota have rarely been reported. In this study, we examined *Sec1* gene expression during mouse development, constructed *Sec1* knockout mice, and sequenced their gut microbial composition. It was found that *Sec1* was expressed at different stages of mouse development. *Sec1* knockout mice have significantly higher intraperitoneal fat accumulation and body weight than wild-type mice. Analysis of gut microbial composition in *Sec1* knockout mice revealed that at the phylum level, *Bacteroidetes* accounted for 68.8%and 68.3% of gut microbial composition in the *Sec1^−^*^/−^ and *Sec1^+^*^/+^ groups, respectively, and *Firmicutes* accounted for 27.1% and 19.7%, respectively; while *Firmicutes*/*Bacteroidetes* were significantly higher in *Sec1^−^*^/−^ mice than in *Sec1^+^*^/+^ mice (39.4% vs. 28.8%). In *verucomicrobia*, it was significantly higher in *Sec1^−^*^/−^ mice than in *Sec1^+^*^/+^ group mice. At the family level, the dominant bacteria *Prevotellaceae*, *Akkermansiaceae*, *Bacteroidaceae*, and *Lacilltobacaceae* were found to be significantly reduced in the gut of *Sec1^−^*^/−^ mice among *Sec1^+^*^/+^ gut microbes, while *Lachnospiraceae*, *Ruminococcaceae*, *Rikenellaceae*, *Helicobacteraceae*, and *Tannerellaceae* were significantly increased. Indicator prediction also revealed the dominant bacteria *Akkermansiaceae* and *Lactobacillaceae* in *Sec1^+^*^/+^ gut microorganisms, while the dominant bacteria *Rikenellaceae*, *Marinifilaceae*, *ClostridialesvadinBB60aceae*, *Erysipelotrichaceae*, *Saccharimonadaceae*, *Clostridiaceae1*, and *Christensenellaceae* in *Sec1^−^*^/−^ group. This study revealed that the *Sec1* gene was expressed in different tissues at different time periods in mice, and *Sec1* knockout mice had significant weight gain, significant abdominal fat accumulation, and significant changes in gut microbial flora abundance and metabolic function, providing a theoretical basis and data support for the study of *Sec1* gene function and effects on gut microbiota-related diseases.

## 1. Introduction

Intestinal microorganisms are an important part of the intestinal mucosal barrier, which is closely related to host digestion, nutrition, metabolism, immunity, and other aspects [1]. It is an “environmental” factor in the human body, and its status and role are equivalent to those of an important “organ” acquired [2]. Absolute and relative concentrations of individual fucosylated oligosaccharides varied widely during one year of lactation, possibly related to hormone-regulated changes in fucosylated intestinal mucin content and effects on infant gut microbiota development [3,4,5]. Different types of milk oligosaccharides are differentially digested by *Bifidobacterium* and *Bacteroides* species and strains. Unlike other types of oligosaccharides, fucosylated oligosaccharides can strongly stimulate key species of mutualistic symbionts [6]. The number of intestinal *Bacteroides* was significantly reduced in people with abnormal glucose metabolism, and the ratio of *Bacteroidetes* to *Firmicutes* was positively correlated with blood glucose levels, indicating that intestinal bacterial changes are closely related to the occurrence of abnormal glucose metabolism [7]. Gill et al. found that the number of intestinal *Firmicutes* increased, *Bacteroides* decreased, and intestinal bacterial diversity was significantly lower in obese people than in normal lean people. An increased *Firmicutes*/*Bacteroidetes* ratio affects not only carbohydrate metabolism but also short-chain fatty acid production (increased acetate production and decreased butyrate production) [8]. The intestinal microecosystem of animals is closely related to the health status, nutrient metabolism, immune function, and disease occurrence in animals [9].

Fucosyltransferases (*FUTs*) are a class of biosynthetic enzymes that are involved in the synthesis of karst oligosaccharides and catalyze the transfer of L-fucose from the donor substrate guanosine diphosphate β-L-fucose to various sugar acceptor substrates, including oligosaccharides, glycoproteins, and glycolipids; fucosylation is a relatively important form of glycosylation modification, which plays an important role in signaling pathways, inflammatory bowel disease, and immune response [10,11]. It is involved in the regulation of Lewis antigens through the glycosylation of Lewis antigenic determinants and can also be involved in the disease process by making terminal glycosylation of other glycoproteins such as transcription factors and receptor proteins [12]. The *Sec1* gene (secretory blood group 1) in mice, also known as *Fut3* and *Fut10*, encodes the galactoside 2-L-fucosyltransferase.

Fucosylation determined by the *FUT2* and *FUT3* genes is an important fucosylation and is critical for mucin maturity and integration function [13,14]. Hypoglycemia of mucin has so far been found in human inflammatory bowel disease and is associated with susceptibility to inflammatory bowel disease [15]; mucin prevents excessive T helper 17 cell responses in murine colitis [16]. *FUT3* can also promote the metastasis of colorectal cancer by initiating epithelial transition through fucosylation modification of TβR (mainly TβRI), affecting p38 signaling and the TGF-β/S-mad pathway [17]. At the same time, down-regulation of *FUT3* reduces the expression of fucosylated antigens and promotes cell adhesion [18,19]. *FUT3* gene polymorphisms have also been revealed to exert an effect on *Helicobacter pylori* infection, and host fucosylated glycoproteins and sialylated glycolipids (Lewis antigens) have been shown to act as pathogenic binding sites for *H. pylori* in the gastric epithelium [20]. In addition to the study of the *FUT3* gene in the process of intestinal diseases, there are also some studies focusing on the effect of *FUT3* on tumor markers, analyzing 22 variants in the FUT gene, and several variants in *FUT2*, *FUT3*, *FUT5*, *FUT6*, and *FUT7* are associated with the risk of intestinal and diffuse gastric cancer and, to a lesser extent, with cardia and non-cardia gastric cancer [21]. Related studies have shown that selection and maintenance of gut microbes are regulated by the *FUT2* and *FUT3* genes encoding fucosyltransferases, and differences in *FUT2* and *FUT3* gene expression result in high individual differences in fucosylated milk oligosaccharide composition in lactating mothers [22].

The expression of several fucosyltransferases involved in Lewis antigen synthesis in gastric cells has been shown to be modulated by the inflammatory cytokines, IL-1b and IL-6, and their prognostic effects on cancer and related motility mechanisms in cancer cells [23]. *Fut3* and Glg1-mediated E-selectin binding activity contribute to the formation of bone metastases in cancer [24]. However, the expression of the *Sec1* gene in various tissues at different growth stages in mice has rarely been reported.

Animal intestinal microorganisms are involved in the regulation of a variety of host metabolic pathways, producing microbial metabolic axes, host and microbial signaling axes, and immunoinflammatory axes that interact with the host, which connect multiple body organs such as the intestine, liver, muscle, and brain [25]. However, there are few reports on the regulatory mechanism of the *Sec1* gene on intestinal microbial bacteria, so this study tried to detect the expression of the *Sec1* gene in various tissues of mice at different stages and constructed a knockout mouse model of the *Sec1* gene to analyze the effect of the *Sec1* gene knockout on intestinal bacteria.

Present in the genomes of prokaryotes such as all bacteria, chlamydia, mycoplasma, rickettsia, spirochetes, and actinomycetes, 16SrDNA is a gene encoding prokaryotic 16SrRNA of about 1500 bp in length and consisting of multiple conserved regions and multiple variable regions [26]. With the mature development of Illumina Hi-Seq technology, 16SrDNA sequencing technology is mostly used in research to reflect the microbial flora under different factors [27].

In this study, we examined *Sec1* expression in various tissues of 1-day, 3, 6, 9, and 12-week-old mice. Body weight and abdominal fat phenotypes were recorded in *Sec1^−^*^/−^ and *Sec1^+^*^/+^ mice, and 16SrDNA sequencing analysis of feces from *Sec1^−^*^/−^ and *Sec1^+^*^/+^ mice was performed in order to understand the effect of *Sec1* knockout on mouse gut microbiota and the relationship with phenotypic differences in mice. By analyzing, the enrichment and diversity of bacteria in the feces of the *Sec1* knockout mice and wild-type mice, the changes in intestinal flora and their effects on individual phenotypes in the *Sec1* knockout mice and wild-type mice were revealed, providing a relevant basis for the study of *Sec1* gene function and mechanism.

## 2. Materials and Methods

### 2.1. Establishment of Knockout Mouse Model

We constructed *Sec1* knockout mice using CRIPSR/Cas9 technology. The construction procedures have been reported in our previous study [28]. Wild-type (C57BL/6J) mice were purchased from the Animal Center of Yangzhou University (Yangzhou, China). The *Sec1* transgenic knockout mice (back handed over to a C57BL/6) were purchased from Cyagen (Guangzhou, China). All mouse husbandry and breeding were performed at the Animal Experimental Center of Yangzhou University (Yangzhou, China). Each mouse was placed in a separate cage and allowed free access to irradiated food and sterile acidified water in a specific pathogen-free facility.

### 2.2. Collection of Feces Samples

*Sec1^−^*^/−^ and *Sec1^+^*^/+^ mice were subjected to fecal collection (N = 13, 12, respectively), and cryovials were numbered and grouped. When collecting feces, warm and wet cotton balls were used to stimulate and gently press the lower abdomen of mice. The fresh feces of mice were collected in the corresponding cryogenic vials and immediately placed in liquid nitrogen. They were transported to the laboratory within 1 h and stored for future use.

### 2.3. Fecal DNA Extraction and PCR Amplification in Sec1^−/−^ versus Sec1^+/+^ Mouse

Fecal samples were thawed and microbial genomic DNA was extracted. PCR amplification conditions for the V3––V4 region: 95 °C for 5 min, then 30 cycles of 95 °C for 1 min, 60 °C for 1 min, 72 °C for 1 min, and then 72 °C for 7 min. The V4 region was expanded (341F: CCTACGGG-NGGCWGCAG; 806R: GGACTACHVGGGTATCTAAT). Amplification system: 50 μL mixture, containing 10 μL 5 × Q5^@^ Reaction Buffer, 10 μL 5× Q5^@^ High GC Enhancer, 1.5 μL 2.5 mM dNTPs, 1.5 μL upstream and downstream primers (10 μM), 0.2 ΜL Q5^@^ High-Fidelity DNA Polymerase, and 50 ng of template DNA. Amplicons were collected in 2% agarose gels and purified with the AxyPrep DNA Gel Extraction Kit (Axygen Biosciences, Union City, CA, USA), and the purified amplicons were sequenced at both ends (PE250) on an Illumina platform.

### 2.4. Library Cconstruction and on-Board Sequencing

Library construction was performed using the library construction kit, and the constructed library was quantified by Qubit quantification and library detection, and double-end sequencing (PE250) was performed on the Illumina platform after qualification.

### 2.5. DNA Extraction and PCR Amplification

Microbial DNA was extracted using the HiPure Soil DNA Kits (or HiPure Stool DNA Kits) (Magen, Guangzhou, China) according to the manufacturer’s protocols. The 16SrDNA V4 region of the ribosomal RNA gene was amplified by PCR (95 °C for 2 min, followed by 27 cycles at 98 °C for 10 s, 62 °C for 30 s, and 68 °C for 30 s, and final extension at 68 °C for 10 min) using primers Arch519F: CAGCMGCCGCGGTAA; Arch915R: GTGCTCCCCCGCCAATTCCT, where the barcode is an eight-base sequence unique to each sample. PCR reactions were performed in triplicate with 50 μL of mixture containing 5 μL of 10 × KOD Buffer, 5 μL of 2.5 mM dNTPs, 1.5 μL of each primer (5 μM), 1 μL of KOD Polymerase, and 100 ng of template DNA.

### 2.6. Illumina Hiseq 2500 Sequencing

Amplicons were extracted from 2% agarose gels and purified using the AxyPrep DNA Gel Extraction Kit (Axygen Biosciences, Union City, CA, USA) according to the manufacturer’s instructions and quantified using the ABI Step One Plus Real-Time PCR System (Life Technologies, Foster City, CA, USA) [29]. Purified amplicons were pooled in equimolar and paired-end sequenced (2 × 250) on an Illumina platform according to the standard protocols.

### 2.7. Sequencing Data Processing

To obtain clean sequence reads, primers, low-quality sequences, and barcode sequences need to be removed. The contralateral clean reads were merged using FLASH (Baltimore, MD, USA, Version 1.2.11) with a minimum overlap of 10 bp and a mismatch error rate of 2%. OTUs are clustered by sequence similarity based on tags (nucleic acid sequences) and divided into different sequence sets (clusters), and a cluster is an operational taxonomic unit (OTU) with 97% similarity using the UPARSE (Version 9.2.64) pipeline to cluster effective tags. OTUs were relatively abundant; 0.1% and were present in less than 1% of the experimental pigs and were removed from further analysis. The most abundant marker sequences were used as representative sequences in each cluster. Venn analysis was performed in the R project Venn Diagram package (Version 1.6.16) and an upset plot was performed in the R project UpSetR package (Version 1.3.3) to identify unique and common OTUs.

### 2.8. Alpha and Beta Diversity Analysis

QIIME (Version 1.9.1) calculated Chao1, Shannon indices, and OTU scarcity curves were generated in the R project ggplot2 package (version 2.2.1) [30]. The Alpha index was compared among groups using a Kruskal-Wallis H test in the R project Vegan package (Version 2.5.3). OTUs scarcity curves were generated in the R project ggplot2 package (version 2.2.1). The Alpha index was compared among groups using a Kruskal-Wallis H test in the R project Vegan package (Version 2.5.3). Microbial gene function was predicted using PICRUSt (Version 2.1.4) software [31]. Predicted genes and their functions were then annotated using the Kyoto Encyclopedia of Genes and Genomes (KEGG) database. Welch’s test was used to calculate the functional difference analysis of the R project Vegam package (Version 2.5.3).

### 2.9. RT-qPCR

The HiScript^®^ Reverse Transcriptase kit (Vazyme Biotech Co., Ltd., Nanjing, China) was used to produce cDNA. Then, qPCR reactions were performed using the SYBR Green master mix (Vazyme Biotech Co., Ltd., Nanjing, China) in an ABI Step-ONE Plus Real-Time PCR System (Applied Biosystems, Foster City, CA, USA). The mouse *GAPDH* was selected as an internal control. Each gene was performed in triplicate, and the relative quantitative of gene expression was calculated using the 2-ΔΔCt method. The primers used for RT-qPCR of *Sec1* are as follows: F: 5′-AAGGATCCAAGCAGTGCTCC-3′; R: 5′-GGGAAGACCACAAGGGATGG-3′.

## 3. Results

### 3.1. Sec1 Expression in Mouse Tissues at Different Time Periods and Phenotypic Differences of Wild Type and Knockout Type Mouse

In order to investigate the expression of the *Sec1* gene in the various tissues of mice, the expression of the *Sec1* gene in the heart, liver, spleen, lung, kidney, and duodenum of 1-day, 3, 6, 9, and 12-week-old mice was detected, and *Sec1* expression was found to be expressed in all tissues of 1-day-old mice, and the expression of *Sec1* in the lung and spleen was relatively high with the increase in the age of the mice. In the intestine, the expression showed a decreasing trend at 1 day, 3 and 6 weeks (Figure 1A–C), increasing at week 9 (Figure 1D), and decreasing at week 12 (Figure 1E).

The body weight of mice at 6, 12, and 18 weeks of age (N = 13, 12, respectively) was followed up and recorded, and it was found that the body weight of mice in the *Sec1^−^*^/−^ group was higher than that in the *Sec1^+^*^/+^ group (*p* < 0.05), and the body weight of mice in the *Sec1^−^*^/−^ group was significantly higher than that in the *Sec1^+^*^/+^ group at 12 and 18 weeks of age (*p* < 0.01) (Figure 1F). Combined with the results of quantification and body weight, 9-week-old *Sec1^+^*^/+^ and *Sec1^−^*^/−^ mice were selected for necropsy, and it was found that *Sec1^−^*^/−^ mice had significantly thickened abdominal fat layer accumulation (Figure 1G).

### 3.2. Effect of Sec1 Knockout on Microbial Species Richness in the Mouse Intestine

The dilution curves of the *Sec1^−^*^/−^ and *Sec1^+^*^/+^ groups were used to evaluate whether the sequencing volume was sufficient and thus indirectly to compare the richness of the species. As shown by the chao1 index, *Sec1^−^*^/−^ was 763.9 and *Sec1^+^*^/+^ was 663.0 (Figure 2A), indicating that the species richness of samples in *Sec1^−^*^/−^ group was higher than that in the *Sec1^+^*^/+^ group; the Shannon index showed that the diversity of microorganisms was *Sec1^−^*^/−^ 6.4 and *Sec1^+^*^/+^ 5.1, respectively (Figure 2B), indicating that the species uniformity of samples in *Sec1^−^*^/−^ and *Sec1^+^*^/+^ groups was good, the diversity was high, and the sequencing data were reliable.

When tags were about 30,000, sob leveled off (Figure 2C), indicating that the species richness of the two was high. It also indicated that the sequencing data were stable and reliable, and the richness of the *Sec1^−^*^/−^ group was higher than that of the *Sec1^+^*^/+^ group. With the increase of effective sequence sequencing depth, the dilution curve first rises rapidly and then tends to be flat, indicating that the sample sequencing data volume is reasonable and the sequencing quality is good, with certain depth and representativeness. The trend of decreasing abundance gradually leveled off with increasing Rank values, indicating better evenness and richness of *Sec1^−^*^/−^ and *Sec1^+^*^/+^ species (Figure 2D).

The significance analysis of Welch’s *t*-test using the Chao1 and Shannon indexes showed that both the Chao1 index (*p* = 0.000005) (Figure 2E) and the Shannon index (*p* = 0.000002) were significantly different between the groups (Figure 2F) (*p* < 0.01). It can be found that there is a highly significant difference in species diversity between the two groups.

### 3.3. Effect of Sec1 Knockout on the Structure of Microbial Bacteria in the Intestine of Mouse

The Wayne diagram showed that the common OTUs were 37 in *Sec1*^−/−^ and *Sec1*^+/+^ groups and 3 in both *Sec1*^−/−^ and *Sec1*^+/+^ groups at the family level (Figure 3A). For OTU cluster analysis, *Sec1^−^*^/−^ and *Sec1*^+/+^ had shared OTUs of 499, accounting for approximately 80% of all. Additionally, among them, the OTUs exclusive to the *Sec1^−^*^/−^ group were 164, and the OTUs exclusive to the *Sec1*^+/+^ group were 80 (Figure 3B).

Beta diversity was analyzed to reflect differences between different groups. The samples constructed by unweighted group average (UPGMA) reflect the similarities and differences among multiple samples. Using the UPGMA clustering method of bray distance, it can be clearly seen that there is a clear separation of samples between *Sec1^+^*^/+^ and *Sec1^−^*^/−^ groups in OTU classification, indicating that there is a significant difference in bacterial structure between the two groups (Figure 3C). Principal coordinate analysis (PCoA) assesses the degree of explanation of the overall difference in bacterial structure by each axis as a percentage (number in brackets of the axis title), and it is generally better to reach more than 50% of the sum of PCo1 and PCo2, so that we can see that the sequencing results are good. In addition, PCoA was performed based on Bray-Curtis (Figure 3D) distance, respectively, and the results showed that the samples were significantly clustered based on Bray-Curtis distance (ADONIS; *Sec1^−^*^/−^ vs. *Sec1^+^*^/+^, *p* < 0.05). Significant difference between groups.

In order to show the nonlinear structural relationship of ecological data among samples, non-metric multidimensional scaling (NMDS) is used to display the species information contained in the samples in the form of multidimensional space points. The results showed that there were significant differences between different groups of samples, and there were significant differences between the two groups, whether based on Bray (Figure 3E) distance (ANOSIM; *Sec1^−^*^/−^ vs. *Sec1^+^*^/+^, *p* < 0.05).

### 3.4. Effect of Sec1 Knockout on the Composition of Microbial Bacteria in the Mouse Intestine

OTUs were clustered to investigate the structural composition of different bacteria. At the phylum level, *Sec1^−^*^/−^ and *Sec1^+^*^/+^ were analyzed, *Bacteroidetes* accounted for 68.8% and 68.3%, and *Firmicutes* accounted for 27.1% and 19.7%, respectively, and the two accounted for the highest proportion, accounting for about 90% of all. *Firmicutes*/*Bacteroidetes* were about 39.4% in the *Sec1^−^*^/−^ group but 28.8% in the *Sec1^+^*^/+^ group, which was higher in the *Sec1^−^*^/−^ group than in the *Sec1^+^*^/+^ group. In *Verrucomicrobia*, *Sec1^+^*^/+^ accounted for 8.8% and the *Sec1^−^*^/−^ group for 0.1% (Figure 4A).

At the family level, the dominant families of *Sec1^+^*^/+^ were *Prevotellaceae*, *Akkermansiaceae*, *Bacteroidaceae*, and *Lactobacillaceae*, and the dominant families of *Sec1^−^*^/−^ were *Lachnospiraceae*, *Ruminococcaceae*, *Rikenellaceae*, *Helicobacteraceae*, and *Tannerellaceae* (Figure 4B). The distribution of flora was approximately the same at the genus level as at the family level, but was more significantly different at *Akkermansiacea* and *Alistipes* (Figure 4C).

At the species level, the gut microbiota of *Sec1^−^*^/−^ mice was significantly changed compared with *Sec1^+^*^/+^ mice, and *Lactobacillus reuteri* and *Lactobacillus gasseri* were significantly decreased in *Sec1* knockout mice, while *FirmicutesumM10-2*, *Bacteroides caecimuris*, and *Lachnospiraceaebacterium-COE1* were increased compared with *Sec1^+^*^/+^ mice (Figure 4D), which was basically consistent with the results at the family level. In response to this phenomenon, two groups of mouse microbiota markers were predicted at the family and species levels.

In order to further explore the marker differences in the *Sec1^−^*^/−^ and *Sec1*^+/+^ groups, Wilcoxon rank sum test was performed at the level of family and species, and *Lachnospiraceae*, *Ruminococcaceae*, *Rikenellaceae*, and *Marinifilaceae* were higher in the *Sec1^−^*^/−^ group than in the *Sec1^+^*^/+^ group at the level of family (*p* < 0.01); *Bacteroidaceae*, and *Lacillaceae* were higher in the *Sec1*^+/+^ group than in the *Sec1^−^*^/−^ group (*p* < 0.01) (Figure 4E); *PrevotellaceaeUCG-001*, *Lactobacillus* were higher in the *Sec1*^+/+^ group than in the *Sec1^−^*^/−^ group (*p* < 0.01) at the genus level (Figure 4F). At the species level, *Lactobacillus reasseri* and *Lactobacillus gasseri* abundance decreased after *Sec1* gene knockdown, while *Lachnospiraceaebacterium28-4*, *Bacteroidescaecimuris*, and *Lachnospiraceaebuterium-COE1* were higher than in the *Sec1^+^*^/+^ group (Figure 4G).

### 3.5. Changes in Gut Microbiota Function in Sec1 Knockout Mouse

PICRUSt software was used to predict the function of the microbiota based on 16SrDNA sequencing genomes. In order to further investigate the differences in KEGG pathways between different groups, the functional distribution of each group was plotted as a general overview. It can be seen that the number of genes associated with Metabolism ranked first, followed by Genetic Information Processing, Cellular Processes, Environmental Information Processing, Organismal Systems, and Human Diseases, and *Sec1^−^*^/−^ was higher than the *Sec1^+^*^/+^ group (Figure 5A,B). Welch ‘s *t*-test; amino acid metabolism; metabolism of terpenoids and polyketides; lipid metabolism; folding, sorting and degradation; xenobiotic biodegradation; and metabolism were performed at Level 2 (Figure 5C). Functional genes involved in carbohydrate metabolism and lipid metabolism pathways were found to be significantly upregulated in the *Sec1*^−^^/−^ group by PICRUSt functional prediction analysis, while disturbances in carbohydrate metabolism and lipid metabolism aggravate negative energy balance and fat accumulation, which cause weight gain, and preclinical evidence supports the role of gut microbes and their metabolites in behavioral responses associated with mood, social interaction, appetite, and food intake.

## 4. Discussion

The mammalian *Sec1* gene encodes a type II membrane protein involved in the function encoding galactosylglycoside 1,2-fucosyltransferase. In this study, we examined the expression of Sec1 in various tissues of mice at different time periods and recorded *Sec1^+^*^/+^ and *Sec1^−^*^/−^ abdominal fat changes at 6, 12, and 18 weeks of age and at 9 weeks of age, and found that the Sec1 gene was expressed in all tissues of the 1d mouse, and the expression of Sec1 was relatively high in the lung and spleen with increasing mouse age. In the intestine, expression showed a decreasing trend at 1 day, 3 and 6 weeks, increased at 9 weeks, and decreased at 12 weeks.

The body weights of mice at 6, 12 and 18 weeks of age (N = 13, 12, respectively) were recorded, and it was found that the body weights of mice in the *Sec1^−^*^/−^ group were higher than those in the *Sec1^+^*^/+^ group (*p* < 0.05), and the body weights of mice in the *Sec1^−^*^/−^ group were significantly higher than those in the *Sec1^+^*^/+^ group at 12 and 18 weeks of age (*p* < 0.01). Combined with quantitative and body weight results, *Sec1^+^*^/+^ and *Sec1^−^*^/−^ mice at 9 weeks were selected for necropsy, and it was found that the abdominal fat layer accumulated and thickened in *Sec1^−^*^/−^ mice. The *Sec1* gene in mice, also known as *Fut3* and *Fut10*, encodes the galactoside 2-L-fucosyltransferase.

Studies have shown that human *FUT3* encodes α-(1,3/4) fucosyltransferase can be associated with abnormal expression in a variety of gastric lesions, including gastritis, intestinal metaplasia, and gastric cancer, through the effect of fucosylation on intestinal diseases. *FUT3* in pigs is highly expressed mainly on the mucosal surface of intestinal tissues (duodenum and jejunum) and is significantly higher in *Escherichia coli* susceptible individuals than in resistant ones. As a fucosyltransferase, *FUT3* indirectly mediates the glycosphingolipid biosynthesis signaling pathway to regulate *E. coli* resistance. In the digestive tract of mice, galactoside 1,2-fucosyltransferase can catalyze the transfer of fucose to a-sialic acid-gm1 to produce fucosylated-ga1, and the process occurs during weaning and is developmentally regulated [32]. In addition, the *Sec1* gene encodes a type II membrane protein that anchors to the Golgi and controls the final step of α (1, 2) fucosylated bicarbonate production by adding terminal fucose to α (1, 2) linkages, and the biological function of fucosylated carbohydrate products is thought to involve cell adhesion and interaction with microorganisms.

Intestinal microbes are mainly divided into six gates, specifically *Firmicutes*, *Bacteroidetes*, *Proteobacteria*, *Actinobacteria*, *Verrucous Microbes*, and *Fusobacteria* [33]. The number of microorganisms in the human body is about 10 times the number of human cells, ranging from 100 trillion to 1000 trillion, in which the number of genes encoded is more than 100 times that of the body’s own genes [34]. In addition, it has been found that the imbalance of intestinal bacteria is related to a series of chronic diseases, including obesity, diabetes, inflammatory bowel disease, and even colon cancer, and affects the physiological metabolism of the human body through its interaction with environmental conditions [35].

In this study, feces from *Sec1* knockout mice and wild-type mice were used as samples for gut microbial community composition analysis, and the results showed that the *Sec1*^+/+^ and *Sec1^−^*^/−^ groups had the highest proportion of abundance in *Bacteroidetes*, *Verrucomicrobia*, and *Firmicutes*. In *Bacteroidetes*, the abundance of *Sec1*^+/+^ and *Sec1^−^*^/−^ groups was flat, with no significant difference; in *Verrucomicrobia*, the *Sec1*^+/+^ group was higher than the *Sec1^−^*^/−^ group, with significant differences; while in *Firmicutes*, the *Sec1*^+/+^ group was lower than the *Sec1^−^*^/−^ group, with significant differences. Gordon and colleagues analyzed 16SrRNA gene sequences of 5088 gut terminal microorganisms from genetically obese mice and lean mice with the same diet and found that the abundance of *Bacteroidetes* bacteria in the gut of obese mice decreased by 50%, while the proportion of *Firmicutes* bacteria increased [36].

This is consistent with the results of human research experiments in which 12 obese patients adhered to a carbohydrate-reducing hypocaloric diet for 1 year and had a significant decrease in body weight, their gut had a reduced abundance ratio of *Firmicutes* to *Bacteroidetes* microorganisms. Ley et al. found that obese subjects had a higher proportion of intestinal *Firmicutes* to *Bacteroidetes* abundance than lean subjects, and with obese/lean mice with intestinal microorganisms similar to the proportion of bacteria abundance gained 20% of body weight after transplantation into germ-free mice [37]. At present, the abundance ratio of *Firmicutes* to *Bacteroidetes* is often used to explain the etiology of obesity and related complications, and the phenomenon has been widely confirmed [38,39].

The dominant families of *Sec1^+^*^/+^ were *Prevotellaceae*, *Akkermansiaceae*, *Bacteroidaceae*, *Lactobacillaceae*, and the dominant families of *Sec1^−^*^/−^ were *Lachnospiraceae*, *Ruminococcaceae*, *Rikenellaceae*, *Helicobacteraceae*, and *Tannerellaceae*, which were significantly different. At the species level, the dominant flora of *Sec1^+^*^/+^ was *Lactobacillus reuteri*, *Lactobacillus gasseri*, and the dominant flora of *Sec1^−^*^/−^ was *Bacteroidescaecimuris*, *Lachnospiraceaebacterium-COE1*, *Lachnospiraceaebacterium-28-4*, and *Clostridiumleptum*. *Akkermansiaceae* (belonging to the phylum Verrucous Microorganisms) inhabiting the human gut are promising new generations of probiotics that play important roles in diseases such as diabetes, obesity, and alcoholic liver disease [40]. In the experiment by Everard et al., *Akkermansia* content was reduced 100-fold in the cecum of mice fed a diet for 8 weeks. *Akkermansia* has beneficial effects on glucose metabolism and intestinal permeability, which are inversely associated with T2DM and have been tested as probiotics in preclinical trials. *Akkermansia muciniphila* administered obesity and type 2 diabetes in mouse models showed that diet-induced metabolic disorders, such as increased fat content and adipose tissue inflammation, were restored [41]. *A. muciniphila*, the most studied member of the *Akkermansiaceae*, has probiotic properties and is usually found to be inversely associated with human diseases such as obesity, diabetes, inflammation, and metabolic disorders [42,43,44]. Bian et al. demonstrated that administration of *A. muciniphila* can improve intestinal inflammation in mice subjected to DSS-induced ulcerative colitis by protecting the intestinal barrier, reducing the levels of inflammatory cytokines, and improving the diversity of intestinal bacteria through microbe-host interactions [45]. Alternatively, mucin acts as the sole source of carbon, nitrogen, and energy for *Akkermansia*, producing acetate, ethanol, propionate, and sulfate during glycolysis, the metabolites propionic acid and acetic acid play an important protective role in the development of obesity and type 2 diabetes [46].

*Lactobacillus* is the dominant bacterial species in the gut of humans and animals and is widely used as a probiotic [47]. *Lactobacillus* has immunomodulatory effects and promotes intestinal host defense through interactions with the immune system [48]. Obesity is associated with a reduction of *Bacteroides* in the gut, while abundant *Bacteroidaceae* may favor good health. There was no significant difference between *Sec1*^+/+^ and *Sec1^−^*^/−^ *Bacteroidetes* at the level of the phylum, but *Bacteroidaceae* became the dominant group in the *Sec1*^+/+^ group at the level of the family, indicating that intestinal bacteria changed after *Sec1* knockout mouse. Short-chain fatty acids (SCFAs) associated with obesity regulation include lactate, succinate, propionate, and butyrate, which are mainly secreted by *Bacteroidetes* bacteria [49]. At the species level, the dominant flora of *Sec1^+^*^/+^ was *Lactobacillus reuteri* and *Lactobacillus gasseri*, of which *Lactobacillus reuteri* had a strong inhibitory effect on TNF-α induced IL-8 expression in human intestinal epithelial cells and could significantly reduce the content of IL-6 [50]. *Prevotellaceae* was previously associated with the degradation and utilization of resistant starch and the production of beneficial SCFAs in the gastrointestinal tract of ruminants and humans [51]. Earlier studies have shown that butyrate can inhibit the progression of non-alcoholic fatty liver disease (NAFLD) by activating the adenosine monophosphate-activated protein kinase (AMPK) signaling pathway, thereby regulating lipid and energy metabolism, insulin sensitivity, and oxidative stress [52]. Other studies have shown that short-chain fatty acids inhibit intestinal inflammation by upregulating the G protein-coupled receptor 43 (GPR43), which allows the effect of hepatic portal pulsatile inflammatory factors. It has been shown that in the rat small intestine, terminal modifications of glycoconjugates change significantly during weaning, with a decrease in sialyltransferase activity and a concomitant increase in fucosyltransferase activity.

In summary, *Sec1* gene expression was lower in the intestine, but phenotypically, *Sec1^−^*^/−^ mice weighed significantly higher than concurrent and *Sec1^+^*^/+^ mice, and abdominal fat accumulation was more pronounced in *Sec1^−^*^/−^ mice. After *Sec1* knockout, Bacteroidaceae, *Akkermansia*, *Lactobacillus reuteri*, *Lactobacillus gasseri*, and other beneficial bacteriaceae related to fatty acid metabolism and immunity were significantly decreased in mice, and it can be inferred that the *Sec1* gene may play an important role in the process of mice growth and development. Knockout mice have altered gut microbiota and reduced numbers of beneficial microbiota, and *Sec1* knockout may cause disturbances in fat metabolism and immune function, which cause related diseases leading to weight gain.

## 5. Conclusions

*Sec1* was expressed in all tissues of 1d mice, and the expression of *Sec1* was relatively high in the lungs and spleen with increasing mouse age. In the intestine, expression showed a decreasing trend at 1 day, 3, and 6 weeks, increased at 9 weeks, and decreased at 12 weeks. *Sec1^−^*^/−^ mice weighed significantly more than concurrent and *Sec1^+^*^/+^ mice. We found that *Bacteroidaceae*, *Akkermansia*, *Lactobacillus reuteri*, *Lactobacillus gasseri*, and other beneficial bacteriaceae related to fatty acid metabolism and immunity were significantly decreased in mice after *Sec1* knockout, and we could infer that *Sec1* knockout resulted in changes in the gut microbiota of mice, and the amount of beneficial bacterial flora was decreased, which may cause disturbances in related metabolism and thus lead to an increase in body weight.

## Figures and Tables

**Figure 1 genes-13-01858-f001:**
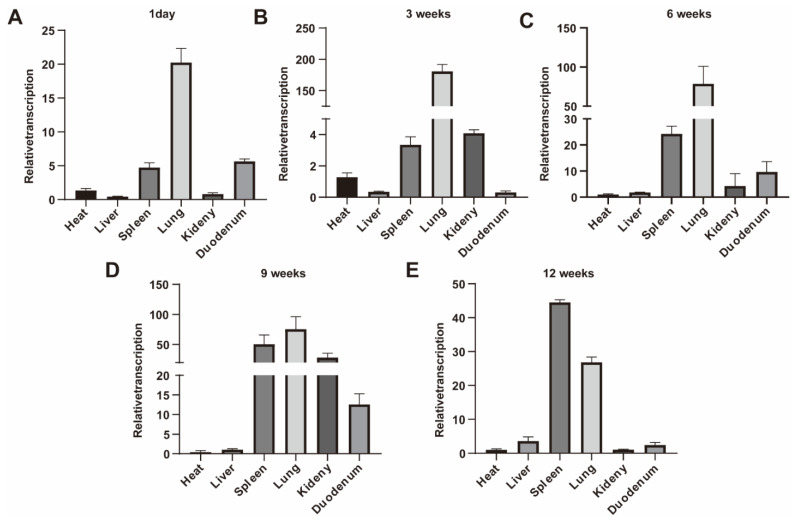
*Sec1* Tissue Expression Profiles and body weight and abdominal fat accumulation in *Sec1^−^*^/−^ and *Sec1^+^*^/+^ mice. (**A**–**E**) *Sec1* gene expression on the heart, liver, spleen, lung, kidney, and duodenum of *Sec1^+^*^/+^ mouse at 1 day, 3, 6, 9, and 12 weeks of age (*n* = 3). (**F**) Body Weight Difference between *Sec1^−^*^/−^ and *Sec1^+^*^/+^ mice. (**G**) Abdominal fat accumulation in *Sec1^−^*^/−^ and *Sec1^+^*^/+^ mouse (Red circle). * *p* < 0.05; ** *p* < 0.01; compared with *Sec1^+^*^/+^ mouse. Data represented mean ± SD.

**Figure 2 genes-13-01858-f002:**
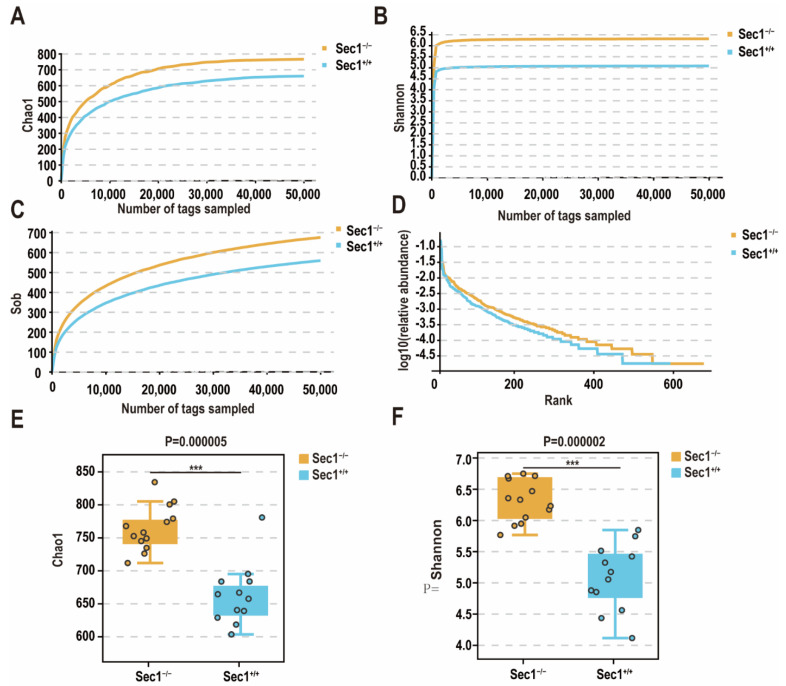
*Sec1^−^*^/−^ mice elevate microbial species richness in the gut. (**A**) The α diversity index is mainly concerned with the species richness of the sample. (**B**) The Shannon index in α diversity index comprehensively reflects the richness and uniformity of the species, so the level of the index is affected by the uniformity, and the more uniform the species distribution in the sample, the greater the diversity. (**C**) Sob dilution curve indirectly reflects the richness of species in the sample. (**D**) The rank abundance curve can visually reflect the taxonomic richness and uniformity included in the sample. In the horizontal direction, the taxonomic abundance is reflected by the width of the curve, the higher the taxonomic richness, the span of the curve on the horizontal axis; the smoothness of the curve in the vertical direction reflects the taxonomic uniformity in the sample, the flatter the curve, the more uniform the species distribution. (**E**) The α diversity is tested for hypothesis, the Chao1 index Welch’s *t* test. (**F**) the Shannon index Welch’s *t* test. *** *p* < 0.001.

**Figure 3 genes-13-01858-f003:**
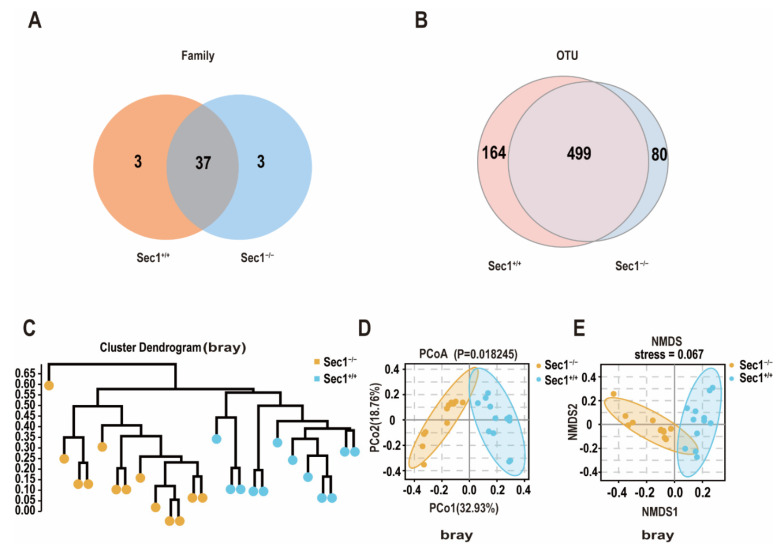
*Sec1* knockout impacts gut microbial architecture and drives diverse microbial responses. (**A**,**B**) Collection of Veen Graph Bacteria Distribution at family and OTU levels in *Sec1^−^*^/−^ and *Sec1*^+/+^ mice (N = 13, 12, respectively). (**C**) Gut microbiomes were clustered by similarity using the UPGMA clustering algorithm on the unweighted UniFrac distance. Each terminal branch represents a sample. Generally, the samples of the same group are clustered into a large branch, and different groups constitute different branches. The longitudinal axis is the distance scale. Samples in the same branch indicate that the bacterial structure is more similar. (**D**) Principal coordinate analysis of multivariate statistical analysis was performed, Bray distance parameter, OTU level for analysis. (**E**) Non-metric multidimensional scale analysis of multivariate statistical analysis, Bray distance parameter, OTU level for analysis. Multivariate statistical analysis is to use the idea of dimensionality reduction to transform the distance matrix between two samples into the figure of a two-dimensional plane and display the difference in bacterial structure through the distribution image of sample points on the plane. The closer the sample distribution on the plane is, the more similar the population structure of the sample is.

**Figure 4 genes-13-01858-f004:**
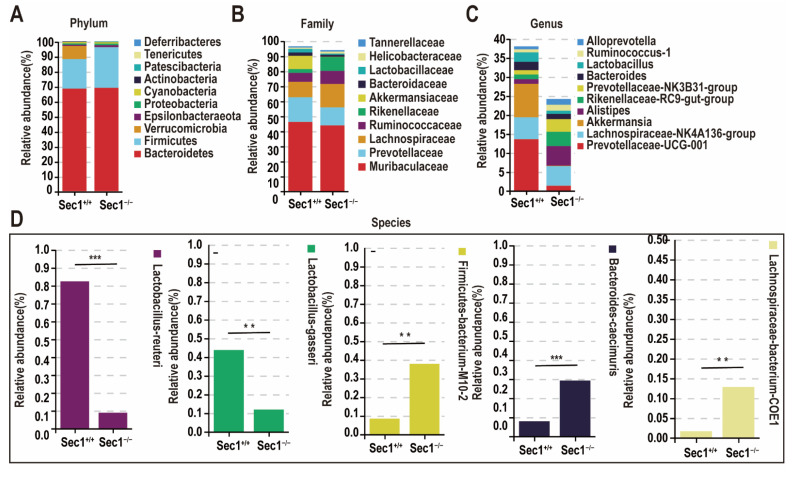
Effect of *Sec1* on gut microbial composition. (**A**–**D**) At the phylum, family, genus, and species levels, the composition of *Sec1^−^*^/−^ and *Sec1^+^*^/+^ bacteria groups. (**E**) Welch’s *t*-test of markers at the family and species levels in *Sec1^−^*^/−^ and *Sec1*^+/+^ mice, whether there was a significant difference in the mean abundance of family between the two groups when the two groups were compared. (**F**) Welch ‘s *t* test for genus-level markers, species level for *Sec1^−^*^/−^ and *Sec1^+^*^/+^ mice, and whether there were significant differences in genus-level mean abundances between the two groups when compared. (**G**) Welch’s *t*-test was performed for the analysis of differences in two groups of species (door-to-species level, filtering for species with a sum of abundances below 0.1% in all samples). ** *p* < 0.01; and *** *p* < 0.001.

**Figure 5 genes-13-01858-f005:**
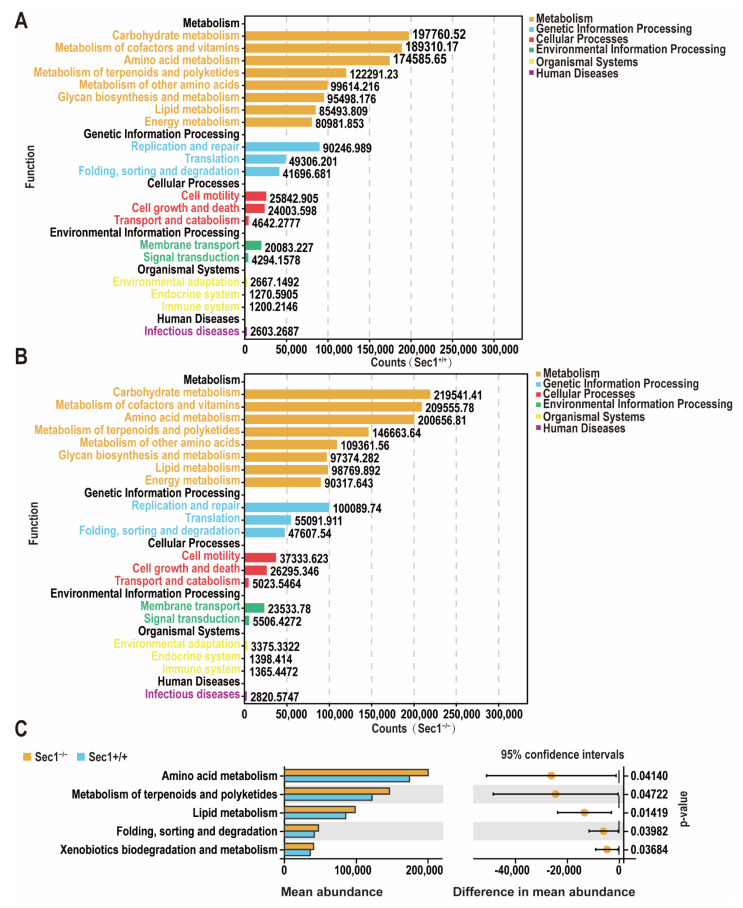
*Sec1* knockout impacts gut microbial community changes and drives diverse functions. (**A**,**B**) *Sec1^−^*^/−^ and *Sec1^+^*^/+^ functional distribution overview. (**C**) The mean abundance of metabolic pathways enriched by the gut microbes in the two groups. Welch ‘s *t* test was used to test the differences at level 2 between the two groups.

## Data Availability

The data presented in this study are available on request from the corresponding author.

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
