# Peer review of "Expression Analysis and the Roles of the *Sec1* Gene in Regulating the Composition of Mouse Gut Microbiota"

_genes, 2022, doi:10.3390/genes13101858_

Round 1

Reviewer 1 Report

Authors should address the following comments.

Line 50: Hypoglycemia of Mucin I... (needs revision, meaning is not clear)

Line 107-115: Very long and complex, needs revision

Justification of time points chosen, 1 day and 3,6,9, and 12 weeks. to be clearly stated.

Figure 4. Genus level data is missing. It is important to include the genus-level data.

How were the species-level data generated, and how were they validated?

Authors have used fecal samples; why not cecal content, which is a better representation? Please justify.

Author Response

Response to Reviewer 1 Comments

Point 1: Line 50: Hypoglycemia of Mucin I... (needs revision, meaning is not clear).

Response 1: Thank you for your comment. And according to your suggestion, we now added further discussion in our revised manuscript.

 “Fucosylation determined by the FUT2 and FUT3 genes is important fucosylation and is critical for mucin maturity and integration function. Hypoglycemia of mucin has so far been found in human inflammatory bowel disease and is associated with susceptibility to inflammatory bowel disease; mucin prevents excessive T helper 17 cell responses in murine colitis.”  (Line 67-72)

Point 2: Line 107-115: Very long and complex, needs revision.

Response 2: Thanks for your comments. And according to your suggestion, we now renewed this sentence in our revised manuscript. “In this study, The Sec1 expression in various tissues of 1 day, 3 weeks, 6, 9and 12 weeks mice were examined, and the body weight and abdominal fat phenotypes were recorded in Sec1−/− and Sec1+/+ mice, and 16SrDNA sequencing analysis of feces from Sec1−/− and Sec1+/+ mice was performed in order to understand the effect of Sec1 knockout on mouse gut microbiota and the relationship with phenotypic differences in mice. By analysing the enrichment and diversity of bacteria in feces of Sec1 knockout mice and wild-type mice, the differential of intestinal flora and their effects on individual phenotypes in Sec1 knockout mice and wild-type mice were revealed, which would provide a relevant basis for the study of Sec1 gene function and mechanism.” (Line 125-132)

Point 3: Justification of time points chosen, 1 day and 3,6,9, and 12 weeks. to be clearly stated.

Response 3: Thanks for your comments. In order to analysis the Sec1 expression in different period, so we chose 1 day and 3, 6, 9, and 12 weeks C57 mice. According to the growth cycle and development of C57 mice, we know that they were weaned at 3 weeks of age, began to mature in late puberty at 6 weeks of age, and sexually mature at 9-12 weeks of age. So, select a more typical time period for recording phenotypes, detection indicators, more representative.

Point 4: Figure 4. Genus level data is missing. It is important to include the genus-level data.

Response 4: Thank you very much for your insightful suggestion. Following your suggestion, we include genus level data in Figure 4 (4C,4F).

Point 5: How were the species-level data generated, and how were they validated?

Response 5: Thank you very much for your comment. Abundance statistics for each species classification were shown using Krona [1] (version 2.6). Species abundance stacking plots were presented using the R language ggplot2 package [2] (version 2.2.1). Species abundance chord maps were drawn using circos [3] (version 0.69-3) software, i.e., presented in a circular layout. Indicatorvalue for each species in each group compared. Statistical tests were also performed using cross-validation to obtain P values (P < 0.05)

[1] Ondov, Brian D., Nicholas H. Bergman, and Adam M. Phillippy. Interactive metagenomic visualization in a Web browser. BMC bioinformatics 12.1 (2011): 385.

[2] Wickham H, Chang W. ggplot2: An implementation of the Grammar of Graphics[J]. R package version 0.7, URL: http://CRAN. R-project. org/package= ggplot2, 2008, 3.

[3] Krzywinski M, Schein J, Birol I, et al. Circos: an information aesthetic for comparative genomics[J]. Genome research, 2009, 19(9): 1639-1645.

Point 6: Authors have used fecal samples; why not cecal content, which is a better representation? Please justify.

Response 6: Thank you very much for your comment. The sampling time was selected at 8-9 am. In order to reduce the influence of external environment, the mouse was selected to press the lower abdomen of the mouse continuously with a warm cotton swab in the super clean bench to stimulate the mouse to defecate. Feces were then loaded into sterilized 1.5ml EP tubes, cryopreserved in liquid nitrogen, and transferred to a -80 ° C freezer for cryopreservation.

 In addition, due to the particularity of transgenic mice, the expansion of transgenic mice in offspring requires that slaughter sampling from the cecum cannot be performed.

Reviewer 2 Report

Ren et al used previously-constructed Sec1-/-  mice to define gut microbiota composition relative to wild-type Sec1+/+ mice using the standard 16SrRNA sequencing method. The study provided sufficient evidence indicating that deficiency of Sec1 altered gut microbiota composition with notable increases in the presence of Firmicutes at phylum level and Rikenellaceae at family level,  and decreased presence of Lactobacillus_reuteri and L. gasseri at species level, possibly affecting nutrition status and fat deposition. Authors may wish to make revisions to improve manuscript.

Specific comments

1)     Reorganization of introduction to first discuss gut microbiota, and FUT genes, in human health and diseases followed by discussion of gut microbiota and Sec1 gene in mouse models would allow readers to better understand as why it is necessary to carry out the current study.

2)      Figure 1: Number of mice used for each group in each panel (time point) should be specified in the legend (number of animals or samples should be disclosed in the legend of all figures!). The variable Y axis scales in each panel are misleading. Using interrupted Y scale for panels B and C are OK. For panels A, D and E, continuous scale from 0 to 150 should be used to remove possible misleading in data interpretation.

3)     Figure 3. Panels A and B the circles are too big relative to elements in other panels. Also in panel B the size of 80 and 164 are the same, which is misleading.

4)     Figure 4 presented the most important data of this study documenting gut microbiota differences between Sec1-/- and Sec1+/+ mice at Phylum, Family and Species levels. However, data presented in Figure 4E at Species level are very much limited not sufficiently coherent with data presented in panel D. For example, the almost 6% more Rikenellaceae in Sec1-/- mice (~9% in Sec1-/- mice  relative to 3% in Sec1+/+ mice) was not reflected in panel E. It is recommended, if possible, that authors revise panel E to present microbiota differences at Species level using a heat map format listing more species than the five currently listed. A heatmap would be better to show overall differences.

5)     Discussion should be focused on the observed differences between Sec1-/- and Sec1+/+ mice at Phylum (Firmicutes), Family (Rikenellaceae) and Species (Lactobacillus_reuteri etc) levels relevant to previously published work. The current discussion is somewhat “diffused”. Discussion should also involve a possible explanation/speculation as why Sec1 deficiency will alter gut microbiota, under the condition that Sec1 gene expression is low in gut tissue (Figure 1).

6)     Despite referring to their previous paper, authors should provide strain and background information regarding mice used in the current study in the method section. Were the Sec1-/- and Sec1+/+ mice used in the current study from same litters? Also animal husbandry, diet, and water conditions should be described as these are important elements affecting gut microbiota. It is assumed that Sec1-/- and Sec1+/+ are housed in the same facility of SPF specification. A general animal age range should also be specified in method, despite that animal age has been clearly stated in some figures.     

Author Response

Response to Reviewer 2 Comments

Point 1: Reorganization of introduction to first discuss gut microbiota, and FUT genes, in human health and diseases followed by discussion of gut microbiota and Sec1 gene in mouse models would allow readers to better understand as why it is necessary to carry out the current study.

Response 1: Thank you very much for your comment. Fut3 gene, also known as Sec1 in mice, also encodes fucosyltransferase, and previous groups have explored the functional mechanism of this gene on pigs and found that it is associated with intestinal function, and constructed Sec1 knockout mouse models. On this basis, the phenotypes of knockout mice at different stages were recorded, and it was found that the mice after knockout had increased body weight and significant abdominal fat accumulation. According to this phenomenon, we explored and analyzed the differences in gut microbiota between knockout and wild-type mice at the same time period. To provide a rationale for the possible effects of the Sec1 gene in mammalian gut microbiota and related diseases. According to your suggestion, we have adjusted the introduction

“1. Introduction

Intestinal microorganism is an important part of intestinal mucosal barrier, which is closely related to host digestion, nutrition, metabolism, immunity and other aspects [1]. It is an "environmental" factor into the human body, and its status and role are equivalent to an important "organ" acquired [2]. Absolute and relative concentrations of individual fucosylated oligosaccharides varied widely during one year of lactation, possibly related to hormone-regulated changes in fucosylated intestinal mucin content and effects on infant gut microbiota development [3-5]. Different types of milk oligosaccharides are differentially digested by Bifidobacterium and Bacteroides species and strains. Unlike other types of oligosaccharides, fucosylated oligosaccharides can strongly stimulate key species of mutualistic symbionts [6]. The number of intestinal Bacteroides was significantly reduced in people with abnormal glucose metabolism, and the ratio of Bacteroidetes to Firmicutes was positively correlated with blood glucose levels, indicating that intestinal bacteria changes are closely related to the occurrence of abnormal glucose metabolism [7]. Gill et al found that the number of intestinal Firmicutes increased, Bacteroides decreased, and intestinal bacterial diversity was significantly lower in obese people than in normal lean people. Increased Firmicutes/Bacteroidetes ratio affects not only carbohydrate metabolism but also short-chain fatty acid production (increased acetate production and decreased butyrate production) [8]. The intestinal microecosystem of animals is closely related to the health status, nutrient metabolism, immune function and disease occurrence of animals [9].

Fucosyltransferases (FUTs) are a class of biosynthetic enzymes that are involved in the synthesis of karst oligosaccharides and catalyze the transfer of L-fucose from the donor substrate guanosine diphosphate β-L-fucose to various sugar acceptor substrates, including oligosaccharides, glycoproteins, and glycolipids; fucosylation is a relatively important form of glycosylation modification, which plays an important role in signaling pathways, inflammatory bowel disease, and immune response [10, 11]. It is involved in the regulation of Lewis antigens through the glycosylation of Lewis antigenic determinants, and can also be involved in the disease process by making terminal glycosylation of other glycoproteins such as transcription factors and receptor proteins [12]. The Sec1 gene (secretory blood group 1) in mouse, also known as Fut3 and Fut10, encodes the galactoside 2-L-fucosyltransferase.

Fucosylation determined by the FUT2 and FUT3 genes is important fucosylation and is critical for mucin maturity and integration function. [13, 14]. Hypoglycemia of mucin has so far been found in human inflammatory bowel disease and is associated with susceptibility to inflammatory bowel disease; [15] mucin prevents excessive T helper 17 cell responses in murine colitis [16]. FUT3 can also promote the metastasis of colorectal cancer by initiating epithelial transition through fucosylation modification of TβR (mainly TβRI), affecting p38 signaling and TGF-β/S-mad pathway [17]. At the same time, down-regulation of FUT3 reduces the expression of fucosylated antigen and promotes cell adhesion [18, 19]. FUT3 gene polymorphisms have also been revealed to exert an effect on H. pylori infection, and host fucosylated glycoproteins and sialylated glycolipids (Lewis antigens) have been shown to act as pathogenic binding sites for H. pylori in the gastric epithelium [20]. In addition, the study of FUT3 gene in the process of intestinal diseases, there are also some studies focusing on the effect of FUT3 on tumor markers, analyzing 22 variants in the FUT gene, and several variants in FUT2, FUT3, FUT5, FUT6 and FUT7 are associated with the risk of intestinal and diffuse gastric cancer and to a lesser extent with cardia and non-cardia gastric cancer [21]. Related studies have shown that selection and maintenance of gut microbes are regulated by the FUT2 and FUT3 genes encoding fucosyltransferases, and differences in FUT2 and FUT3 gene expression result in high individual differences in fucosylated milk oligosaccharide composition in lactating mothers [22].

The expression of several fucosyltransferases involved in Lewis antigen synthesis in gastric cells has been shown to be modulated by the inflammatory cytokines IL-1b and IL-6, prognostic effects on cancer and related motility mechanisms in cancer cells [23]. Fut3 and Glg1-mediated E-selectin binding activity contributes to the formation of bone metastases in cancer [24]. However, the expression of Sec1 gene in various tissues at different growth stages in mouse has rarely been reported.

Animal intestinal microorganisms are involved in the regulation of a variety of host metabolic pathways, producing microbial metabolic axes, host and microbial signaling axes and immunoinflammatory axes that interact with the host, which connect multiple body organs such as the intestine, liver, muscle and brain [25]. However, there are few reports on the regulatory mechanism of Sec1 gene on intestinal microbial bacteria, so this study tried to detect the expression of Sec1 gene in various tissues of mouse at different stages, and constructed a knockout mouse model of Sec1 gene to analyze the effect of Sec1 gene knockout on intestinal bacteria.

16SrDNA is a gene encoding prokaryotic 16SrRNA, about 1500 bp in length, which is present in the genomes of prokaryotes such as all bacteria, chlamydia, mycoplasma, rickettsia, spirochetes and actinomycetes, and consists of multiple conserved regions and multiple variable regions [26]. With the mature development of Illumina Hi-Seq technology, 16SrDNA sequencing technology is mostly used in research to reflect the microbial flora under different factors [27].”

Point 2: Figure 1: Number of mice used for each group in each panel (time point) should be specified in the legend (number of animals or samples should be disclosed in the legend of all figures!). The variable Y axis scales in each panel are misleading. Using interrupted Y scale for panels B and C are OK. For panels A, D and E, continuous scale from 0 to 150 should be used to remove possible misleading in data interpretation.

Response 2: Thank you very much for your comment. According to your suggestion, we added the sample number "n = 3" for each group of mice to the figure notes in Figure 1. The time period of mice was added directly above the figure. Because the expression of Sec1 gene varies greatly in different tissues at different times, the Y-axis scale uniformly selects 0-150, which will not intuitively reflect the expression of Sec1 gene at different times in different tissues.

Point 3:  Figure 3. Panels A and B the circles are too big relative to elements in other panels. Also in panel B the size of 80 and 164 are the same, which is misleading.

Response 3: Thank you very much for your comment. Following your suggestion, we have revised the size of the elements in panels A and B. 

Point 4:  Figure 4 presented the most important data of this study documenting gut microbiota differences between Sec1-/- and Sec1+/+ mice at Phylum, Family and Species levels. However, data presented in Figure 4E at Species level are very much limited not sufficiently coherent with data presented in panel D. For example, the almost 6% more Rikenellaceae in Sec1-/- mice (~9% in Sec1-/- mice relative to 3% in Sec1+/+ mice) was not reflected in panel E. It is recommended, if possible, that authors revise panel E to present microbiota differences at Species level using a heat map format listing more species than the five currently listed. A heatmap would be better to show overall differences.

Response 4: Thank you very much for your comment. Following your suggestion, we added genus-level species distribution stacking plot data (C, F). For species level (G), differential analysis of 2 grouped species (phylum to species level, filtered for species with a sum of abundances below 0.1% in all samples) was performed with Welch 's T-test, and the results were screened at a threshold of P-value < 0.05, and after screening, six differential flora were obtained at species level.

Point 5: Discussion should be focused on the observed differences between Sec1-/- and Sec1+/+ mice at Phylum (Firmicutes), Family (Rikenellaceae) and Species (Lactobacillus_reuteri etc) levels relevant to previously published work. The current discussion is somewhat “diffused”. Discussion should also involve a possible explanation/speculation as why Sec1 deficiency will alter gut microbiota, under the condition that Sec1 gene expression is low in gut tissue (Figure 1).

Response 5: Thank you very much for your comment. In summary, Sec1 gene expression was lower in the intestine, but phenotypically, Sec1−/− mice weighed significantly higher than concurrent and Sec1+/+ mice, and abdominal fat accumulation was more pronounced in Sec1−/− mice. After Sec1 knockout, Bacteroidaceae, Akkermansia, Lactobacillus reuteri, Lactobacillus gasseri and other beneficial bacteriaceae related to fatty acid metabolism and immunity were significantly decreased in mice, and it can be inferred that Sec1 gene may play an important role in the process of mice growth and development. Knockout mice have altered gut microbiota and reduced numbers of beneficial microbiota, and Sec1 knockout may cause disturbances in fat metabolism and immune function, which cause related diseases leading to weight gain. (Line 385-393)

Point 6: Despite referring to their previous paper, authors should provide strain and background information regarding mice used in the current study in the method section. Were the Sec1-/- and Sec1+/+ mice used in the current study from same litters? Also animal husbandry, diet, and water conditions should be described as these are important elements affecting gut microbiota. It is assumed that Sec1-/- and Sec1+/+ are housed in the same facility of SPF specification. A general animal age range should also be specified in method, despite that animal age has been clearly stated in some figures.   

Response 6: Thank you very much for your comment.  Wild-type (C57BL/6J) mice were purchased from the Animal Center of Yangzhou University (Yangzhou, Jiangsu, China). Sec1 transgenic knockout mice (back handed over to a C57BL/6) were purchased from Cyagen (Guangzhou, China). All mouse husbandry and breeding were performed at the Animal Experimental Center of Yangzhou University (Yangzhou, China). Each mouse was placed in a separate cage and allowed free access to irradiated food and sterile acidified water in a specific pathogen-free facility. (Line 397-402)
